# Sample Complexity for Quadratic Bandits: Hessian Dependent Bounds and Optimal Algorithms

**Qian Yu**
Princeton University
qy4628@princeton.edu

**Yining Wang**
University of Texas at Dallas
yining.wang@utdallas.edu

**Baihe Huang**
University of California, Berkeley
baihe_huang@berkeley.edu

**Qi Lei**
New York University
ql518@nyu.edu

**Jason D. Lee**
Princeton University
Jasondl@princeton.edu

## Abstract

In stochastic zeroth-order optimization, a problem of practical relevance is understanding how to fully exploit the local geometry of the underlying objective function. We consider a fundamental setting in which the objective function is quadratic, and provide the first tight characterization of the optimal Hessian-dependent sample complexity. Our contribution is twofold. First, from an information-theoretic point of view, we prove tight lower bounds on Hessian-dependent complexities by introducing a concept called *energy allocation*, which captures the interaction between the searching algorithm and the geometry of objective functions. A matching upper bound is obtained by solving the optimal energy spectrum. Then, algorithmically, we show the existence of a Hessian-independent algorithm that universally achieves the asymptotic optimal sample complexities for all Hessian instances. The optimal sample complexities achieved by our algorithm remain valid for heavy-tailed noise distributions, which are enabled by a truncation method.

## 1 Introduction

Stochastic optimization in the zeroth order (gradient-free) setting has attracted significant attention in recent decades. It naturally arises in various applications such as autonomous driving [12], AI gaming [22], robotics [14], healthcare [23], and education [13]. This setting is particularly important when the gradient of the objective function cannot be directly evaluated or is expensive.

An important case of interest in the zeroth order setting is the bandit optimization of smooth and strongly-convex functions. Although there are ample results regarding the minimax rates of this problem [1, 8, 21, 6, 3, 24], little is known about how its complexity depends on the geometry of the objective function $f$ near the global optimum $x^*$, as specified by the quadratic approximation $\frac{1}{2}(x - x^*)^\top \nabla^2 f(x^*)(x - x^*)$. As an initial step, we investigate the following natural questions:

- For zeorth-order bandit optimization problems of quadratic functions of the form $\frac{1}{2}(x - x_0)^\top A(x - x_0)$, what is the optimal instance-dependent upper bound with respect to $A$?

- Is there an algorithm that universally achieves the optimal instance-dependent bounds for all quadratic functions, but without the knowledge of Hessian?

As our main contributions, we fully addressed the above questions as follows. First, we established the tight Hessian-dependent upper and lower bounds of the simple regret. Our bounds indicate asymptotic sample complexity bounds of $\text{Tr}^2(A^{-\frac{1}{2}})/(2\epsilon)$ to achieve $\epsilon$ accuracy. This covers the

37th Conference on Neural Information Processing Systems (NeurIPS 2023).

minimax lower bound of $\Omega(d^2/\epsilon)$ in [21]. Second, we prove the existence of a Hessian-independent algorithm with a matching upper bound of $O\left(\mathrm{Tr}^2(A^{-\frac{1}{2}})/\epsilon\right)$. Thus, we complete the theory of zeroth-order bandit optimization on quadratic functions.

**Related works beyond linear or convex bandit optimization.** Compared to their linear/convex counterparts, much less is known for finding optimal bandit under nonlinear/non-convex reward function. A natural next step beyond linear bandits is to look at quadratic reward functions, as studied in this paper and some prior work (see reference on bandit PCA [15, 4], its rank-1 special cases [18, 11], bilinear [10], low-rank linear [19] and some other settings [9, 5, 16]).

When getting beyond quadratic losses, prior work on non-convex settings mainly focuses on finding achieving $\epsilon$-stationary points instead of $\epsilon$-optimal reward [17], except for certain specific settings (see, e.g., [7, 25, 26, 20, 26, 20]).

## 2 Problem Formulation and Main Results

We first present a rigorous formulation for the stochastic zeroth-order optimization problem studied in this paper. Given a fixed dimension parameter $d$, let $f : \mathcal{B} \to \mathbb{R}$ be an unknown objective function defined on the closed unit $L_2$-ball $\mathcal{B}$ of the Euclidean space $\mathbb{R}^d$, which takes the following form for some positive-semidefinite (PSD) matrix $A$.

$$f(\boldsymbol{x}) = \frac{1}{2}(\boldsymbol{x} - \boldsymbol{x}_0)^{\mathsf{T}} A (\boldsymbol{x} - \boldsymbol{x}_0). \tag{1}$$

For each time $t \in [T]$, an optimization algorithm $\mathcal{A}$ produces a query point $\boldsymbol{x}_t \in \mathcal{B}$, and receives

$$y_t = f(\boldsymbol{x}_t) + w_t. \tag{2}$$

The algorithm can be *adaptive*, so that $\mathcal{A}$ is described by a sequence of conditional distributions for choosing each $\boldsymbol{x}_t$ based on all prior observations $\{\boldsymbol{x}_\tau, y_\tau\}_{\tau < t}$. Then we only assume that the noises $\{w_t\}_{t=1}^{T-1}$ are independent random variables with zero mean and unit variance, i.e., $\mathbb{E}[w_t|\boldsymbol{x}_t] = 0$ and $\mathbb{E}[w_t^2|\boldsymbol{x}_t] \leq 1$.

For brevity, we use $\mathcal{F}(A)$ to denote all functions $f$ satisfying equation (1) for some $\boldsymbol{x}_0 \in \mathcal{B}$. We are interested in the following minimax *simple* regret for any PSD matrix $A$.

$$\mathfrak{R}(T; A) := \inf_{\mathcal{A}} \sup_{f \in \mathcal{F}(A)} \mathbb{E}\left[f(\boldsymbol{x}_T)\right]. \tag{3}$$

The above quantity characterizes the simple regrets achievable by algorithms with perfect Hessian information. We also aim to identify the existence of algorithms that universally achieves the minimax regret for all $A$, without having access to that knowledge.

Our first result provides a tight characterization for the asymptotics of the minimax regret.

**Theorem 2.1.** *For any PSD matrix $A$, we have*

$$\limsup_{T \to \infty} \mathfrak{R}(T; A) \cdot T \leq \frac{1}{2}\left(\mathrm{Tr}(A^{-\frac{1}{2}})\right)^2, \tag{4}$$

*where $A^{-\frac{1}{2}}$ denotes the pseudo inverse of $A^{\frac{1}{2}}$. Moreover, if the distributions of $w_t$ are i.i.d. standard Gaussian, the above bound provides a tight characterization, i.e., there is a matching lower bound that the following equality is implied.*

$$\lim_{T \to \infty} \mathfrak{R}(T; A) \cdot T = \frac{1}{2}\left(\mathrm{Tr}(A^{-\frac{1}{2}})\right)^2.$$

We prove Theorem 2.1 in Section 3. More generally, we also provide a full characterization of $\mathfrak{R}(T; A)$ for the non-asymptotic regime, stated as follows and proved in Appendix C.

**Theorem 2.2.** *For any PSD matrix $A$ and $T > 3\dim A$, where $\dim A$ denotes the rank of $A$, we have*

$$\mathfrak{R}(T; A) = \begin{cases} \Theta\left(\dfrac{\left(\sum_{k=1}^{k^*} \lambda_k^{-\frac{1}{2}}\right)^2}{T} + \lambda_{k^*+1}\right) & \text{if } k^* < \dim A \\ \Theta\left(\dfrac{(\mathrm{Tr}(A^{-\frac{1}{2}}))^2}{T}\right) & \text{otherwise} \end{cases}$$

where $\lambda_j$ is the $j$th smallest eigenvalue of $A$ and $k^*$ is the largest integer in $\{0, 1, ..., \dim A\}$ satisfying $T \geq \left(\sum_{k=1}^{k^*} \lambda_k^{-\frac{1}{2}}\right)\left(\sum_{k=1}^{k^*} \lambda_k^{-\frac{3}{2}}\right)$.

**Remark 2.3.** *While our formulation requires the algorithm to return estimates from the bounded domain, i.e., $\boldsymbol{x}_T \in \mathcal{B}$, our lower bound applies to algorithms without this constraint as well. This can be proved by showing that the worst-case simple regret over all $f \in \mathcal{F}(A)$ is always achievable by estimators satisfying $\boldsymbol{x}_T \in \mathcal{B}$. Their construction can be obtained using the projection step in Algorithm 1, and proof details can be found in Appendix A.*

Finally, we provide a positive answer to the existence of universally optimal algorithms, stated in the following Theorem. We present the algorithm and prove its achievability guarantee in Section 4.

**Theorem 2.4.** *There exists an algorithm $\mathcal{A}$, which does not depend on the Hessian parameter $A$, such that for $A$ being any PSD matrix, the achieved minimax simple regret satisfies*

$$\limsup_{T \to \infty} \sup_{f \in \mathcal{F}(A)} \mathbb{E}\left[f(\boldsymbol{x}_T)\right] \cdot T = O\left((\mathrm{Tr}(A^{-\frac{1}{2}}))^2\right).$$

## 3 Proof of Theorem 2.1

To motivate the main construction ideas, we start by proving a weaker version of the lower bound in Theorem 2.1. We use the provided intuition to construct a matching upper bound. Then we complete the proof by strengthening the lower bound through a Bayes analysis and the uncertainty principle.

### 3.1 Proof of a Weakened Lower Bound

Assuming $A$ being positive definite, we prove the following inequality when the additive noises $w_t$'s are i.i.d. standard Gaussian.

$$\liminf_{T \to \infty} \mathfrak{R}(T; A) \cdot T \geq \Omega\left((\mathrm{Tr}(A^{-\frac{1}{2}}))^2\right). \tag{5}$$

Throughout this section, we fix any basis such that $A = \mathrm{diag}(\lambda_1, ..., \lambda_d)$. We construct a class of hard instances by letting

$$\boldsymbol{x}_0 \in \mathcal{X}_\mathrm{H} \triangleq \left\{ (x_1, x_2, ..., x_d) \,\middle|\, x_k = \pm\sqrt{\frac{\lambda_k^{-\frac{3}{2}}\left(\sum_{j=1}^d \lambda_j^{-\frac{1}{2}}\right)}{2T}}, \forall k \in [d] \right\}.$$

We investigate a Bayes setting where the objective function is defined by equation (1) and $\boldsymbol{x}_0$ is uniformly random on the above set. For convenience, let $\boldsymbol{x}_t = (x_{1,t}, ..., x_{d,t})$ be the action of the algorithm at time $t$. We define the *energy spectrum* to be a vector $\boldsymbol{R} = (R_1, ..., R_d)$ with each entry given by

$$R_k = \mathbb{E}\left[\sum_{t=1}^T x_{k,t}^2\right].$$

Intuitively, our proof is to show that an allocation of at least $R_k \geq \Omega(\lambda_k^{-2} x_k^{-2})$ energy is required to correctly estimate each $x_k$ with $\Theta(1)$ probability. Note that for any entry that is incorrectly estimated, a penalty of $\Omega(\lambda_k x_k^2)$ is applied to the simple regret. This penalty is proportional to the required energy, which is due to the design of each $x_k$. Meanwhile, the total expected energy is upper bounded by $T$, which is no greater than the summation of the individual requirements. Therefore, an $\Omega(1)$ fraction of the penalty is guaranteed, resulting in an overall effect of $\Omega\left(\sum_k \lambda_k x_k^2\right) = \Omega\left((\mathrm{Tr}(A^{-\frac{1}{2}}))^2/T\right)$ on the simple regret.

Rigorously, consider any fixed algorithm, we define the estimation cost of each entry as follows.

$$E_k = \mathbb{E}\left[\frac{1}{2}\lambda_k(x_{k,T} - x_k)^2\right].$$

For brevity, let $s_k = \text{sign}(x_k)$. The above function is minimized by the minimum mean square error (MMSE) estimator (e.g., see [2, Sec. 4.6]), which depends on the following log-likelihood ratio (LLR) function.

$$L_k \triangleq \log \frac{\mathbb{P}\left[s_k = 1 | \{\boldsymbol{x}_\tau, y_\tau\}_{\tau < T}\right]}{\mathbb{P}\left[s_k = -1 | \{\boldsymbol{x}_\tau, y_\tau\}_{\tau < T}\right]}.$$

Specifically, the MMSE estimator is given by $\widehat{x}_{k,T} \triangleq |x_k| \tanh \frac{L_k}{2}$, and the resulting error conditioned on any fixed $\{\boldsymbol{x}_\tau, y_\tau\}_{\tau < T}$ has an expectation of $x_k^2 \cdot \text{sech}^2 \frac{L_k}{2}$. Hence, we have the following lower bound for each cost entry.

$$E_k \geq \mathbb{E}\left[\frac{1}{2}\lambda_k(\widehat{x}_{k,T} - x_k)^2\right] = \frac{1}{2}\lambda_k x_k^2 \cdot \mathbb{E}\left[\text{sech}^2 \frac{L_k}{2}\right]. \tag{6}$$

By the Gaussian noise assumption, the conditional expectation of LLR can be written as follows.

$$\mathbb{E}\left[L_k | s_k\right] = \mathbb{E}\left[\sum_t -\frac{\left(y_t - \frac{1}{2}\lambda_k(x_{k,t} - x_k)^2\right)^2}{2} + \frac{\left(y_t - \frac{1}{2}\lambda_k(x_{k,t} + x_k)^2\right)^2}{2}\bigg| s_k\right]$$

$$= 2s_k\left(\lambda_k x_k\right)^2 \mathbb{E}\left[\sum_t x_{k,t}^2 \bigg| s_k\right].$$

The above quantity equals the KL divergence between the distributions of action-reward sequences generated by $s_k = \pm 1$. Clearly, it has the following connection to the energy spectrum.

$$\mathbb{E}\left[L_k s_k\right] = 2\left(\lambda_k x_k\right)^2 \mathbb{E}\left[\sum_t x_{k,t}^2\right] = 2\left(\lambda_k x_k\right)^2 R_k. \tag{7}$$

Recall inequality (6), it also implies the following lower bound of the mean-squared error.

$$\mathbb{E}\left[L_k s_k\right] = \mathbb{E}\left[L_k \mathbb{E}\left[s_k | L_k\right]\right] = \mathbb{E}\left[L_k \tanh \frac{L_k}{2}\right] \geq 2 - 2\mathbb{E}\left[\text{sech}^2 \frac{L_k}{2}\right] \geq 2 - \frac{4E_k}{\lambda_k x_k^2}. \tag{8}$$

Combine inequality (7) and (8), the overall simple regret can be bounded as follows.

$$\mathbb{E}\left[f(\boldsymbol{x}_T)\right] = \sum_k E_k \geq \frac{1}{2}\sum_k \left(1 - \lambda_k^2 x_k^2 R_k\right)\lambda_k x_k^2. \tag{9}$$

Note that the definition of $\mathcal{X}_H$ implies that the value of each $x_k^2$ is fixed. Hence,

$$\mathbb{E}\left[f(\boldsymbol{x}_T)\right] \geq \frac{\left(\sum_j \lambda_j^{-\frac{1}{2}}\right)^2}{4T} - \frac{\left(\sum_j \lambda_j^{-\frac{1}{2}}\right)^2}{8T^2}\sum_k R_k \geq \frac{\left(\sum_j \lambda_j^{-\frac{1}{2}}\right)^2}{8T} = \frac{\left(\text{Tr}(A^{-\frac{1}{2}})\right)^2}{8T},$$

where the last inequality uses the fact that the total energy is upper bounded by the number of samples.

Finally, when $T$ is sufficiently large, we have $\|\boldsymbol{x}_0\|_2 \leq 1$ almost surely, which implies that our hard-instance functions belong to the set of objective functions $\mathcal{F}(A)$. Therefore, $\mathbb{E}\left[f(\boldsymbol{x}_T)\right]$ provides a lower bound of the asymptotic minimax regret, and we can conclude that

$$\liminf_{T \to \infty} \Re(T; A) \cdot T \geq \frac{1}{8}\left(\text{Tr}(A^{-\frac{1}{2}})\right)^2.$$

**Remark 3.1.** *The validity of the hard-instance functions requires $T = \Omega\left(\left(\sum_k \lambda_k^{-\frac{1}{2}}\right)\left(\sum_k \lambda_k^{-\frac{3}{2}}\right)\right)$, which is consistent with the transition threshold in Theorem 2.2 to the non-asymptotic regimes.*

### 3.2 Proof of the Upper Bound

Now we provide a proof for equation (4), which is implied by the following result.

**Proposition 3.2.** *For any PSD matrix $A$, let $k^*$ be the rank, $\lambda_1, ..., \lambda_{k^*}$ be the non-zero eigenvalues, and $\boldsymbol{e}_1, \boldsymbol{e}_2, ..., \boldsymbol{e}_{k^*}$ be the associated orthonormal eigenvectors. Then the expected simple regret achieved by algorithm 1 satisfies*

$$\limsup_{T \to \infty} \sup_{f \in \mathcal{F}(A)} \mathbb{E}[f(\boldsymbol{x}_T)] \cdot T \leq \frac{1}{2}\left(\text{Tr}(A^{-\frac{1}{2}})\right)^2. \tag{10}$$

---

**Algorithm 1** Hessian Dependent Algorithm

---

**procedure** HESSIAN DEPENDENT ESTIMATION($\lambda_1, \lambda_2, ..., \lambda_{k^*}, \boldsymbol{e}_1, ..., \boldsymbol{e}_{k^*}, T$)
  **for** $k \leftarrow 1$ to $k^*$ **do**
    Let $R_k = \dfrac{\lambda_k^{-\frac{1}{2}}}{\sum_{j=1}^{k^*} \lambda_j^{-\frac{1}{2}}} \cdot (T - 2d - 1), t_k = \lceil R_k/2 \rceil$.
    Let $\alpha_k = -\frac{1}{2\lambda_k}\left(\text{Sample}(\boldsymbol{e}_k, t_k) - \text{Sample}(-\boldsymbol{e}_k, t_k)\right)$.  ▷ Obtain an unbounded estimator
  **end for**
    **return** $\boldsymbol{x}_T = \text{argmin}_{\boldsymbol{x} \in \mathcal{B}} \sum_{k=1}^{k^*} \lambda_k (\alpha_k - \boldsymbol{x} \cdot \boldsymbol{e}_k)^2$     ▷ Projection to $\mathcal{B}$
**end procedure**

**procedure** SAMPLE($\boldsymbol{x}, t$)
    **return** the average of $t$ samples of $f$ at $\boldsymbol{x}$
**end procedure**

---

*Proof.* Consider an eigenbasis of $A$ with $\boldsymbol{e}_1, ..., \boldsymbol{e}_{k^*}$ being the first $k^*$ vectors. For each $k \in [k^*]$, the algorithm allocates $R_k$ energy to estimate the $k$th entry of $\boldsymbol{x}_0$. The value of $R_k$ is chosen such that the hard instances in the earlier subsection maximizes (modulo a constant factor) the lower bound in inequality (9) (i.e., $R_k$ satisfies $x_k^2 \asymp 1/(\lambda_k^2 R_k)$) while ensuring the total number of samples does not exceed $T - 1$. By the zero-mean and unit-variance assumptions of the noise distribution, we have

$$\mathbb{E}[(\alpha_k - \boldsymbol{x}_0 \cdot \boldsymbol{e}_k)^2] = \frac{1}{2\lambda_k^2 t_k} \leq \frac{\lambda_k^{-\frac{3}{2}} \cdot \sum_{j=1}^{k^*} \lambda_j^{-\frac{1}{2}}}{T - 2d - 1}.$$

This essentially provides an unbounded estimator $\widehat{\boldsymbol{x}} \triangleq \sum_{k=1}^{k^*} \alpha_k \boldsymbol{e}_k$ that satisfies

$$\limsup_{T \to \infty} \sup_{f \in \mathcal{F}(A)} \mathbb{E}[f(\widehat{\boldsymbol{x}})] \cdot T = \limsup_{T \to \infty} \sup_{\boldsymbol{x}_0 \in \mathcal{B}} \frac{1}{2} \sum_{k=1}^{k^*} \lambda_k \mathbb{E}[(\alpha_k - \boldsymbol{x}_0 \cdot \boldsymbol{e}_k)^2] \cdot T$$

$$\leq \frac{\left(\sum_{j=1}^{k^*} \lambda_j^{-\frac{1}{2}}\right)^2}{2} \cdot \limsup_{T \to \infty} \frac{T}{T - 2d - 1}$$

$$= \frac{1}{2}\left(\text{Tr}(A^{-\frac{1}{2}})\right)^2. \tag{11}$$

Then, to obtain $\boldsymbol{x}_T$ within the bounded constraint set, $\widehat{\boldsymbol{x}}$ is projected to $\mathcal{B}$ under a pseudometric defined by matrix $A$. Due to the convexity of set $\mathcal{B}$, we have $f(\widehat{\boldsymbol{x}}) \geq f(\boldsymbol{x}_T)$ with probability 1 (see Appendix A for a proof). Therefore, inequality (10) is implied by inequality (11). □

### 3.3 Proof of the Lower Bound with Tight Constant Factors

To complete the proof of Theorem 2.1, it remains to show that for standard Gaussian noise, we have

$$\liminf_{T \to \infty} \mathfrak{R}(T; A) \cdot T \geq \frac{1}{2}\left(\text{Tr}(A^{-\frac{1}{2}})\right)^2. \tag{12}$$

Notice that the object function and reward feedbacks depend only on the projections of $\boldsymbol{x}_0$ and query points onto the column (or row) space of $A$. Hence, it suffices to focus on algorithms with actions that are constrained on this subspace. This reduces the original problem to an equivalent instance that is defined on a (possibly) lower dimensional space and by a non-singular $A$. Therefore, we only need to prove inequality (12) for those reduced cases (i.e., when $A$ is full-rank).

We lower bound the minimax regret by comparing them with a Bayes estimation error, where $\boldsymbol{x}_0$ has a prior distribution that violates the bounded-norm constraint. Formally, for any fixed algorithm $\mathcal{A}$, we can consider its expected simple regret $\mathbb{E}[f(\boldsymbol{x}_T)]$ over an extended class of objective functions where $f$ is defined by equation (1) but $\boldsymbol{x}_0$ is chosen from the entire Euclidian space. Then for any distribution of $\boldsymbol{x}_0$, the overall expectation is upper bounded by the following inequality.

$$\mathbb{E}_{\boldsymbol{x}_0}\left[\mathbb{E}[f(\boldsymbol{x}_T)]\right] \leq \mathbb{P}[\boldsymbol{x}_0 \in \mathcal{B}] \cdot \sup_{f \in \mathcal{F}(A)} \mathbb{E}[f(\boldsymbol{x}_T)] + \mathbb{E}\left[\mathbb{1}(\boldsymbol{x}_0 \notin \mathcal{B}) \cdot \sup_{\boldsymbol{x} \in \mathcal{B}} f(\boldsymbol{x})\right], \tag{13}$$

where the first term on the RHS above is obtained by taking the supremum over all objective functions that satisfies $x_0 \in \mathcal{B}$, then the second term is obtained from the adversarial choice over all estimators for $x_0 \notin \mathcal{B}$. Compare the RHS of inequality (13) with equation (3), a lower bound of the minimax simple regret $\mathfrak{R}(T; \mathcal{A})$ can be obtained by taking the infimum over algorithm $\mathcal{A}$ on both sides. Hence, it remains to characterize the optimal Bayes estimation error on the LHS.

To provide a concrete analysis, let $x_0$ be a Gaussian vector with zero mean and a covariance of $T^{-\frac{2}{3}} \cdot I_d$, where $I_d$ denotes the identity matrix.[1] Under this setting, conditioned on any realization of queries $x_1, ..., x_{T-1}$ and feedbacks $y_1, ..., y_{T-1}$, the posterior distribution of $x_0$ is proportional to

$$\exp\left(-\frac{\|x_0\|_2^2 \cdot T^{\frac{2}{3}}}{2} - \sum_{t=1}^{T-1} \frac{(y_t - \frac{1}{2}(x_t - x_0)^\mathsf{T} A(x_t - x_0))^2}{2}\right).$$

Clearly, the Bayesian error can be lower bounded by the expectation of the conditional covariance, which is further characterized by the following principle (see Appendix B for a proof).

**Proposition 3.3** (Uncertainty Principle). *Let $Z$ be a random variable on any measurable space and $\theta$ be a real-valued random variable dependent of $Z$. If the conditional distribution of $\theta$ given $Z$ has a density function $f_Z(\theta)$, and $\ln f_Z(\theta)$ has a second derivative that is integrable over $f_Z$, then*

$$\mathbb{E}\left[\mathrm{Var}[\theta|Z]\right] \geq \frac{1}{\mathbb{E}\left[-\frac{\partial^2}{\partial \theta^2} \ln f_Z(\theta)\right]}.$$

Hence, by taking the second derivative, the squared estimation error for the $k$th entry of $x_0$ is lower bounded by the inverse of the following expectation.

$$\mathbb{E}\left[T^{\frac{2}{3}} + \left(\sum_{t=1}^{T-1} A(x_t - x_0)(x_t - x_0)^\mathsf{T} A^\mathsf{T} - Aw_t\right)_{kk}\right]$$

$$= T^{\frac{2}{3}} + \mathbb{E}\left[\left(\sum_{t=1}^{T-1} A(x_t - x_0)(x_t - x_0)^\mathsf{T} A^\mathsf{T}\right)_{kk}\right],$$

where $(\cdot)_{kk}$ denotes the $k$th diagonal entry of the given matrix. Recall that we chose a basis where $A$ is diagonal. The overall Bayes error is bounded as follows

$$\mathbb{E}_{x_0}\left[\mathbb{E}[f(x_T)]\right] \geq \sum_k \frac{\lambda_k}{2} \Big/ \left(T^{\frac{2}{3}} + \lambda_k^2 \mathbb{E}\left[\left(\sum_{t=1}^{T-1}(x_t - x_0)(x_t - x_0)^\mathsf{T}\right)_{kk}\right]\right),$$

where $\lambda_k = (A)_{kk}$ is the $k$th eigenvalue of $A$. By Cauchy's inequality, the RHS above can be further bounded by

$$\frac{\frac{1}{2}\left(\sum_k \lambda_k^{-\frac{1}{2}}\right)^2}{\sum_k \left(T^{\frac{2}{3}}\lambda_k^{-2} + \mathbb{E}\left[\left(\sum_{t=1}^{T-1}(x_t - x_0)(x_t - x_0)^\mathsf{T}\right)_{kk}\right]\right)} = \frac{\frac{1}{2}\left(\mathrm{Tr}(A^{-\frac{1}{2}})\right)^2}{T^{\frac{2}{3}}\mathrm{Tr}(A^{-2}) + \sum_{t=1}^{T-1}\mathbb{E}\left[\|x_t - x_0\|_2^2\right]}.$$

Note that by triangle inequality

$$\mathbb{E}\left[\|x_t - x_0\|_2^2\right] \leq \mathbb{E}\left[(1 + \|x_0\|_2)^2\right] \leq \left(1 + \mathbb{E}\left[\|x_0\|_2^2\right]^{\frac{1}{2}}\right)^2 = \left(1 + d^{\frac{1}{2}}T^{-\frac{1}{3}}\right)^2,$$

where $d$ is the dimension of the action space. We have obtained a lower bound of $\mathbb{E}_{x_0}\left[\mathbb{E}[f(x_T)]\right]$ that is independent of the algorithm. Formally,

$$\inf_{\mathcal{A}} \mathbb{E}_{x_0}\left[\mathbb{E}[f(x_T)]\right] \geq \frac{1}{2}\left(\mathrm{Tr}(A^{-\frac{1}{2}})\right)^2 \Big/ \left(T\left(1 + d^{\frac{1}{2}}T^{-\frac{1}{3}}\right)^2 + T^{\frac{2}{3}}\mathrm{Tr}(A^{-2})\right).$$

We apply the above estimate to inequality (13). By taking the infimum over all algorithms,

$$\liminf_{T \to \infty} \mathfrak{R}(T; \mathcal{A}) \cdot T \geq \liminf_{T \to \infty} \frac{\inf_{\mathcal{A}} \mathbb{E}_{x_0}\left[\mathbb{E}[f(x_T)]\right] - \mathbb{E}\left[\mathbb{1}(x_0 \notin \mathcal{B}) \cdot \sup_{x \in \mathcal{B}} f(x)\right]}{\mathbb{P}[x_0 \in \mathcal{B}]} \cdot T$$

---

[1] For the purpose of our proof, the covariance of $x_0$ can be arbitrary as long as their inverse is asymptotically large but sublinear w.r.t. $T$.

Observe that for our Gaussian prior,

$$\lim_{T\to\infty} \mathbb{P}[\boldsymbol{x}_0 \in \mathcal{B}] = 1,$$

$$\lim_{T\to\infty} \mathbb{E}\left[\mathbb{1}(\boldsymbol{x}_0 \notin \mathcal{B}) \cdot \sup_{\boldsymbol{x}\in\mathcal{B}} f(\boldsymbol{x})\right] \cdot T = 0.$$

Hence, all terms above can be replace by closed-form functions. and we have

$$\liminf_{T\to\infty} \mathfrak{R}(T; A) \cdot T \geq \liminf_{T\to\infty} \inf_{\mathcal{A}} \mathbb{E}_{\boldsymbol{x}_0}\left[\mathbb{E}[f(\boldsymbol{x}_T)]\right] \cdot T$$

$$\geq \liminf_{T\to\infty} \frac{1}{2}\left(\mathrm{Tr}(A^{-\frac{1}{2}})\right)^2 \bigg/ \left(\left(1 + d^{\frac{1}{2}}T^{-\frac{1}{3}}\right)^2 + T^{-\frac{1}{3}}\mathrm{Tr}(A^{-2})\right)$$

$$= \frac{1}{2}\left(\mathrm{Tr}(A^{-\frac{1}{2}})\right)^2.$$

## 4 Proof of Theorem 2.4

To prove the universal achievability result, we need to develop a new algorithm to learn and incorporate the needed Hessian information. Particularly, the procedure required for achieving the optimal rates is beyond simply adding an initial stage with an arbitrary Hessian estimator, for there are two challenges. First, any Hessian estimation algorithm would result in a mean squared error of $\Omega(1/T)$ in the worst case, which translates to a cost of $\Omega(1/T)$ in the minimax simple regret. This induced cost often introduces a multiplicative factor that is order-wise larger than the desired $O\left((\mathrm{Tr}(A^{-\frac{1}{2}}))^2\right)$. Second, to utilize any estimated Hessian, the analysis for the subsequent stages often requires the estimation error to have a light-tail distribution, which is not guaranteed for general linear estimators when the additive noise in the observation model has a heavy-tail distribution.

To overcome these challenges, we present two main algorithmic building blocks in the following subsections. The first uses $o(T)$ samples to obtain a rough estimate of the Hessian, and achieves low-error guarantees with high probability through the introduction of a truncation method. The second estimates the global minimum with carefully designed sample points to minimize the error contributed from the Hessian estimation. We show that this sample phase can be written in the form of a two-step descent. Finally, we show the combination of two stages provides an algorithm that proves Theorem 2.4.

### 4.1 Initial Hessian Estimate and Truncation Method

Consider any sufficiently large $T$, we first use $T_0 = \lceil T^{0.8} \rceil$ samples to find a rough estimate of $A$. Particularly, we rely on the following result.

**Proposition 4.1.** *For any fixed dimension $d$, there is an algorithm that samples on $T_0$ predetermined points and returns an estimation $\widehat{A}$, such that for any fixed $\alpha \in (0, 0.5)$, $\beta \in (0, +\infty)$, and PSD matrix $A$,*

$$\lim_{T_0\to\infty} \sup_{f\in\mathcal{F}_A} \mathbb{P}\left[||\widehat{A} - A||_{\mathrm{F}} \geq T_0^{-\alpha}\right] \cdot T_0^{\beta} = 0, \tag{14}$$

*where $||\cdot||_{\mathrm{F}}$ denotes the Frobenius norm.*

*Proof.* We first consider the 1D case. Let $y_+$, $y_-$, and $y$ each be samples of $f$ at $1$, $-1$, and $0$, respectively. When the additive noises are subgaussian, we have that $(y_+ + y_- - 2y)$ is an unbiased estimator of $A$ with an error that is subgaussian. By repeating and averaging over this process $\lfloor T_0/3 \rfloor$ times, the squared estimation error is reduced to $O(1/T_0)$, and the normalized error is subgaussian, which satisfies the statement in the proposition.

However, recall that in Section 2 we did not assume the subgaussianity of $w_t$. A modification of the above estimator is required to achieve the same guanrantee in equation (14). Here we propose a truncation method, which projects each measurement $(y_+ + y_- - 2y)$ to a bounded interval $[-T_0^{0.5}, T_0^{0.5}]$. Specifically, the returned $\widehat{A}$ is the average of $\lfloor T_0/3 \rfloor$ samples of $\max\{\min\{y_+ + y_- - 2y, T_0^{0.5}\}, -T_0^{0.5}\}$, which provides a guaranteed superpolynomial tail bound. A more detailed discussion and analysis for the truncation method is provided in Appendix D.1.

For general $d$, one can return an estimator that satisfies the same light-tail requirement, for example, by applying the above 1D estimator repetitively poly$(d)$ times to obtain estimates of all entries of $A$. Then the overall error probability is controlled by the union bound. $\qquad\square$

## 4.2 Two-Step Descent Stage

Let $\widehat{A}$ be any estimator that satisfies the condition in Proposition 4.1 for $\alpha = 0.4$ and $\beta = 1.6$. Asymptotically, this implies the eigenvalues and eigenvectors of $\widehat{A}$ are close to that of $A$. We choose an eigenbasis of $\widehat{A}$ and remove vectors with vanishingly small eigenvalues to approximate the row space of $A$. Then, we estimate the entries of $\boldsymbol{x}_0$ in these remaining directions following an energy allocation defined based on $\widehat{A}$.

Specifically, consider any fixed realization of $\widehat{A}$, let $\widehat{\lambda}_1, \widehat{\lambda}_2, ...$ be its eigenvalues in the non-increasing order, $\widehat{\boldsymbol{e}}_1, \widehat{\boldsymbol{e}}_2, ...$ be the corresponding eigenvectors, and $k^*$ be the largest integer such that $\widehat{\lambda}_{k^*} \geq T^{-0.2}$. We present the detailed steps in Algorithm 2, where $T_1$ denotes the remaining available samples, i.e., $T - T_0$.

---

**Algorithm 2**

---

    **procedure** QUADRATIC SEARCH$(\widehat{\lambda}_1, \widehat{\lambda}_2, ..., \widehat{\lambda}_{k^*}, \widehat{\boldsymbol{e}}_1, ..., \widehat{\boldsymbol{e}}_{k^*}, T_1)$
        **for** $k \leftarrow 1$ to $k^*$ **do**
            Let $p_k = \dfrac{\widehat{\lambda}_k^{-\frac{1}{2}}}{4\sum_{j=1}^{k^*} \widehat{\lambda}_j^{-\frac{1}{2}}}$, $t_k = \lceil p_k \cdot (T_1 - 4d - 1)\rceil$.
            Let $\alpha_k = -\frac{1}{2\widehat{\lambda}_k}$Truncated Diff$(\widehat{\boldsymbol{e}}_k, -\widehat{\boldsymbol{e}}_k, t_k)$.
        **end for**
        Let $\widetilde{\boldsymbol{x}} = \sum_{k=1}^{k^*} \alpha_k \widehat{\boldsymbol{e}}_k$, $\widehat{\boldsymbol{x}} = \widetilde{\boldsymbol{x}} \cdot \min\{1, 1.5/||\widetilde{\boldsymbol{x}}||_2\}$     ▷ Projecting to a **0**-centered hyperball

        **for** $k \leftarrow 1$ to $k^*$ **do**
            Let $\beta_k = -\frac{4}{\widehat{\lambda}_k}$Truncated Diff$(\frac{\widehat{\boldsymbol{e}}_k + 2\widehat{\boldsymbol{x}}}{4}, \frac{-\widehat{\boldsymbol{e}}_k + 2\widehat{\boldsymbol{x}}}{4}, t_k)$.     ▷ Obtain an unbounded estimator
        **end for**
        **return** $\boldsymbol{x}_T = \text{argmin}_{\boldsymbol{x}=\sum_{k=1}^{k^*} \theta_k \widehat{\boldsymbol{e}}_k \in \mathcal{B}} \sum_{k=1}^{k^*} \widehat{\lambda}_k (\beta_k - \theta_k)^2$     ▷ Projection to $\mathcal{B}$
    **end procedure**

    **procedure** TRUNCATED DIFF$(\boldsymbol{x}_0, \boldsymbol{x}_1, t)$
        **for** $k \leftarrow 1$ to $t$ **do**
            Let $y_+, y_-$ be a sample of $f$ at $\boldsymbol{x}_0, \boldsymbol{x}_1$, respectively
            Compute the projection of the difference $y_+ - y_-$ to the interval $[-t^{0.5}, t^{0.5}]$, i.e., let $z_k = \max\{\min\{y_+ - y_-, t^{0.5}\}, -t^{0.5}\}$
        **end for**
        **return** $\frac{1}{t}\sum_{k=1}^{t} z_k$
    **end procedure**

---

We use the eigenbasis of $\widehat{A}$ and let $\widehat{A}_0$ be the diagonal matrix given by diag$(\widehat{\lambda}_1, \widehat{\lambda}_2, ..., \widehat{\lambda}_{k^*}, 0, ..., 0)$. In the first for-loop, we essentially estimated $A\boldsymbol{x}_0$, and then computed the first $k^*$ entries of its product with the pseudo-inverse of $\widehat{A}_0$ (note that the rest of the entries are all zero). Since $\widehat{A}_0$ is a good estimate of $A$ with high probability, we use it to compute the optimal energy spectrum similar to the instance-dependent case, and allocate the measurements accordingly.

Recall the proof in Section 3. By the analysis of the truncation method in Appendix D.1, the variable $\widehat{\boldsymbol{x}}$ could have served as an estimator of $\boldsymbol{x}_0$ if $\widehat{A}_0 = A$, where the estimation process reduces to the Hessian-dependent case. However, $\widehat{A}_0$ relies only on $O(T^{0.8})$ samples, which generally leads to an expected penalty of $\Theta(T^{-0.8}) = \omega(1/T)$ in simple regret if used in place of $A$. We reserve a fraction of samples for a second for-loop to fine-tune the estimation in order to achieve the optimal Hessian-dependent simple regrets.

### 4.3 Regret Analysis

Recall our assumption on the Hessain Estimator and $T_0 = O(T^{0.8})$. The probability for $||\widehat{A} - A||_F \geq T^{-0.32}$ is $o(1/T)$. As our algorithm always returns $x_T$ with bounded norms, the simple regret contributed by this exceptional case is negligible. Hence, we can focus on instances where $||\widehat{A} - A||_F \leq T^{-0.32}$.

Under such scenario, any non-zero $\widehat{\lambda}_k$ is close to a non-zero eigenvalue of $A$. Specifically, we have $|\widehat{\lambda}_k - \lambda_k| \leq T^{-0.32}$ for all $k$, where each $\lambda_k$ is the $k$th largest eigenvalue of $A$. Recall the definition of $\widehat{A}_0$. It implies that for sufficiently large $T$, $k^*$ equals the rank of $A$, and all non-zero eigenvalues of $\widehat{A}_0$ are bounded away from zero. Consequently, $||\widehat{A}_0^{-1}||_F = ||A^{-1}||_F(1 + o(1))$, which does not scale w.r.t. $T$, where $M^{-1}$ denotes the pseudo-inverse for any symmetric $M$. We also have $(\text{Tr}(\widehat{A}_0^{-\frac{1}{2}}))^2 = O\left((\text{Tr}(A^{-\frac{1}{2}}))^2\right)$.

The closeness between $\widehat{A}$ and $A$ also implies information on the eigenvectors of $\widehat{A}_0$. Recall that $\widehat{A}_0$ is obtained by removing the diagonal entries of $\widehat{A}$ (in their eigenbasis) that are below a threshold $T^{-0.2}$. For sufficiently large $T$, the removed entries are associated with the $d - k^*$ smallest eigenvalues, which are no greater than $T^{-0.32}$. Hence, as a rough estimate, we have $||\widehat{A}_0 - A||_F = o(T^{-0.3})$.

These conditions can be used to characterize the distribution of $\widehat{x}$. As mentioned earlier, from the analysis of the truncation method, each $\alpha_k$ concentrates around $\widehat{e}_k \widehat{A}_0^{-1} A x_0$. Hence, $\widetilde{x}$ concentrates near $\widehat{A}_0^{-1} A x_0$, and the truncation method ensures that

$$\mathbb{E}\left[\left(\widetilde{x} - \widehat{A}_0^{-1} A x_0\right)^\mathsf{T} \widehat{A}_0 \left(\widetilde{x} - \widehat{A}_0^{-1} A x_0\right)\right] = O\left((\text{Tr}(\widehat{A}_0^{-\frac{1}{2}}))^2\right)/T = O\left((\text{Tr}(A^{-\frac{1}{2}}))^2\right)/T, \quad (15)$$

$$\mathbb{P}\left[\left\|\widetilde{x} - \widehat{A}_0^{-1} A x_0\right\|_2 \geq T^{-0.4}\right] = o\left(1/T\right). \quad (16)$$

Note that the $L_2$-norm of $\widehat{A}_0^{-1} A x_0$ is no greater than $1 + o(1)$ under the condition of $||\widehat{A}_0 - A||_F = o(T^{-0.3})$. Formally, by triangle inequality

$$||\widehat{A}_0^{-1} A x_0||_2 \leq ||\widehat{A}_0^{-1} \widehat{A}_0 x_0||_2 + ||\widehat{A}_0^{-1}(\widehat{A}_0 - A) x_0||_2 \leq 1 + ||\widehat{A}_0^{-1}||_F \cdot ||\widehat{A}_0 - A||_F = 1 + o(1).$$

We can apply inequality (16) to show that the $L_2$-norm of $\widetilde{x}$ is no greater than $1 + o(1)$ with $1 - o(1/T)$ probability. Under such high probability cases, the projection of $\widetilde{x}$ to the hyperball of radius 1.5 remains identical. Hence, by the PSD property of $\widehat{A}_0$, which is due to the convergence of its eigenvalues, we can replace all $\widetilde{x}$ in inequality (15) with $\widehat{x}$ and obtain that

$$\mathbb{E}\left[\left(\widehat{x} - \widehat{A}_0^{-1} A x_0\right)^\mathsf{T} \widehat{A}_0 \left(\widehat{x} - \widehat{A}_0^{-1} A x_0\right)\right] = O\left((\text{Tr}(A^{-\frac{1}{2}}))^2\right)/T. \quad (17)$$

The same analysis can also be performed for each $\beta_k$, which concentrates near $\widehat{e}_k \widehat{A}_0^{-1} A(2x_0 - \widehat{x})$. Formally, let $\widetilde{x}_T \triangleq \sum_k \beta_k \widehat{e}_k$, we have

$$\mathbb{E}\left[\left(\widetilde{x}_T - \widehat{A}_0^{-1} A(2x_0 - \widehat{x})\right)^\mathsf{T} \widehat{A}_0 \left(\widetilde{x}_T - \widehat{A}_0^{-1} A(2x_0 - \widehat{x})\right)\right] = O\left((\text{Tr}(A^{-\frac{1}{2}}))^2\right)/T. \quad (18)$$

The above results for the two descent steps can be combined, using triangle inequality and Proposition 4.2 below, to obtain the following inequality (See Appendix D for their proofs).

$$\mathbb{E}\left[(\widetilde{x}_T - z)^\mathsf{T} \widetilde{A}_0 (\widetilde{x}_T - z)\right] = O\left((\text{Tr}(A^{-\frac{1}{2}}))^2\right)/T. \quad (19)$$

where we denote $z \triangleq \left(2\widehat{A}_0^{-1} A - \left(\widehat{A}_0^{-1} A\right)^2\right) x_0$ for brevity.

**Proposition 4.2.** *Let $\widehat{A}_0, Z, y$ be variables dependent on a parameter $T \in \mathbb{N}$, where $\widehat{A}_0, Z$ are PSD matrices and $y$ belongs to the column space of $\widehat{A}_0$. If $\limsup_{T \to \infty} ||\widehat{A}_0^{-1}||_F < \infty$ and $\lim_{T \to \infty} ||Z - \widetilde{A}_0||_F = 0$, then*

$$y^\mathsf{T} Z y \leq (1 + ||Z - \widehat{A}_0||_F ||\widehat{A}_0^{-1}||_F)(y^\mathsf{T} \widehat{A}_0 y) = (1 + o(1))(y^\mathsf{T} \widehat{A}_0 y).$$

Assume the correctness of inequality (19), the remainder of the proof consists of two parts. First, we show that the vector $z$ can be viewed as a Taylor approximation for a projection of $x_0$ onto the column space of $\widehat{A}_0$, hence, $\widetilde{x}_T$ could achieve the needed simple regret. Then, we show that the projection of $\widetilde{x}_T$ to the unit hypersphere to obtain $x_T$ induces negligible cost, so that the validity constraint can be satisfied the same time.

For the first part, let $P_1$ denote the projective map onto the column space of $\widehat{A}_0$, i.e., $P_1 = \widehat{A}_0 \widehat{A}_0^{-1}$ (here and in the following, $\widehat{A}_0^{-1}$ denotes the pseudo inverse of $\widehat{A}_0$). We consider the Hessian of $f$ on this restricted subspace, which is given by $A_1 \triangleq P_1 A P_1$. Note that the definition of $\widehat{A}_0$ implies $\widehat{A}_0 = P_1 \widehat{A} P_1$. We have $||\widehat{A}_0 - A_1||_F = ||P_1(\widehat{A} - A)P_1||_F \leq ||\widehat{A} - A||_F \leq T^{-0.32}$. Then recall that the non-zero eigenvalues of $\widehat{A}_0$ are bounded away from 0, i.e., $||\widehat{A}_0^{-1}||_F = o(T^{0.32})$. We have that $A_1$ is invertible within the column space of $\widehat{A}_0$ (i.e., $A_1 A_1^{-1} = P_1$) for sufficiently large $T$.

The pseudo inversion of $A_1$ provides a point of equivalence to $x_0$ within the column space of $\widehat{A}_0$. Particularly, let $z_0 \triangleq A_1^{-1} A x_0$, we have $P_1 z = z$ and $f(x) = (x - z_0)^\intercal A(x - z_0)$ for any $x \in \mathbb{R}^d$ and sufficiently large $T$. The second equality is due to $A(x_0 - z_0) = (A - AA_1^{-1}A)x_0 = 0$ for sufficiently large $T$, which is implied by the following proposition (see Appendix D.4 for a proof).

**Proposition 4.3.** *For any symmetric matrix $A$ and any symmetric projection map $P_1$ (i.e., $P_1^2 = P_1 = P_1^\intercal$), if $A_1 \triangleq P_1 A P_1$ and $A$ has the same rank, then the pseudo inverse $A_1^{-1}$ satisfies $A = AA_1^{-1}A$.*

We show that the error $z - z_0$ is bounded by $o(T^{-0.5})$. Observe that $P_1 \widehat{A}_0^{-1} = \widehat{A}_0^{-1} P_1 = \widehat{A}_0^{-1}$ and $A_1^{-1} P_1 = A_1^{-1}$, we have

$$z - z_0 = \left( A_1^{-1} A - \left( 2\widehat{A}_0^{-1} A - \left( \widehat{A}_0^{-1} A \right)^2 \right) \right) x_0 = A_1^{-1} \left( \left( \widehat{A}_0 - A_1 \right) \widehat{A}_0^{-1} \right)^2 A x_0.$$

From $||\widehat{A}_0 - A_1||_F \leq T^{-0.32}$, $||\widehat{A}_0^{-1}||_F = ||\widehat{A}^{-1}||_F \cdot O(1)$ and the fact that $A_1$ is restricted to the column space of $\widehat{A}_0$, we can derive that $||A_1^{-1}||_F = ||\widehat{A}^{-1}||_F \cdot O(1)$, which also does not scale w.r.t. $T$. Therefore, the above equality implies that $||z - z_0||_2 = o(T^{-0.5})$. As a consequence, we have following inequalities due to inequality (19) and triangle inequality.

$$\mathbb{E}\left[ (\widetilde{x}_T - z_0)^\intercal \widetilde{A}_0 (\widetilde{x}_T - z_0) \right] = O\left( (\text{Tr}(A^{-\frac{1}{2}}))^2 \right) / T, \tag{20}$$

For the second part, we first note that the L2 norm of $z_0$ is bounded by the spectrum norm of $A_1^{-1} A$. Specifically,

$$||z_0||_2 = ||A_1^{-1} A x_0||_2 \leq ||x_0||_2 \cdot ||A_1^{-1} A|| \leq ||A_1^{-1} A||,$$

where $|| \cdot ||$ denotes the spectrum norm. Recall the definition of $A_1$, we have $A_1^{-1} A A_1 = A_1^{-1} P_1 A P_1 A_1 = A_1^{-1} A_1 A_1$. Hence,

$$||A_1^{-1} A||^2 = ||A_1^{-1} A^2 A_1^{-1}|| = \left|\left| A_1^{-1} \left( A_1 + (A - A_1) \right)^2 A_1^{-1} \right|\right| = \left|\left| P_1 + A_1^{-1} (A - A_1)^2 A_1^{-1} \right|\right|$$

$$\leq ||P_1|| + \left|\left| A_1^{-1} (A - A_1)^2 A_1^{-1} \right|\right|.$$

Note that $||A - A_1||_F \leq ||\widehat{A}_0 - A_1||_F + ||\widehat{A}_0 - A_1||_F \leq o(T^{-0.3})$. By the fact that $||A_1^{-1}||_F = ||\widehat{A}^{-1}||_F \cdot O(1)$, we have $||A_1^{-1} A||^2 \leq 1 + o(T^{-0.6})$, and $||z_0||_2 \leq 1 + o(T^{-0.6})$.

Therefore, we have $\min_{x \in \mathcal{B}} (z_0 - x)^\intercal \widehat{A}_0 (z_0 - x) = o(T^{-1.2})$. The projection of $\widetilde{x}_T$ to the unit hypersphere guarantees the following bound (see Appendix A for a proof).

$$\mathbb{E}\left[ (x_T - z_0)^\intercal \widetilde{A}_0 (x_T - z_0) \right] \leq \mathbb{E}\left[ (\widetilde{x}_T - z_0)^\intercal \widetilde{A}_0 (\widetilde{x}_T - z_0) \right] + o(T^{-1.2}) = O\left( (\text{Tr}(A^{-\frac{1}{2}}))^2 \right) / T.$$

Note that both $x_T$ and $z_0$ belong to the column space of $\widehat{A}_0$. From Proposition 4.2, we can substitute $\widehat{A}_0$ by $A$ for the above inequality and obtain

$$\mathbb{E}[(x_T - z_0)^\intercal A(x_T - z_0)] = O\left( (\text{Tr}(A^{-\frac{1}{2}}))^2 \right) / T. \tag{21}$$

Then the theorem is proved from the fact that $f(x_T) = (x_T - z_0)^\intercal A(x_T - z_0)$.

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

# A  Projection Lemma

**Proposition A.1.** *For any PSD matrix $A$ with dimension $d$, any closed convex set $\mathcal{B}$ in the Euclidian space $\mathbb{R}^d$, and $\widehat{\boldsymbol{x}} \in \mathbb{R}^d$, let*

$$\boldsymbol{x}^* = \underset{\boldsymbol{x} \in \mathcal{B}}{\operatorname{argmin}}\, g(\widehat{\boldsymbol{x}}, \boldsymbol{x})$$

*where*

$$g(\boldsymbol{u}, \boldsymbol{v}) \triangleq (\boldsymbol{u} - \boldsymbol{v})^{\mathsf{T}} A(\boldsymbol{u} - \boldsymbol{v}),$$

*then*

$$g(\boldsymbol{x}^*, \boldsymbol{x}_0) \le g(\widehat{\boldsymbol{x}}, \boldsymbol{x}_0) \qquad \forall \boldsymbol{x}_0 \in \mathcal{B}.$$

*More generally,*

$$g(\boldsymbol{x}^*, \boldsymbol{z}_0) \le g(\widehat{\boldsymbol{x}}, \boldsymbol{z}_0) + \min_{\boldsymbol{x} \in \mathcal{B}} g(\boldsymbol{z}_0, \boldsymbol{x}) \qquad \forall \boldsymbol{z}_0 \in \mathbb{R}^d.$$

*Proof.* We first provide a proof for $\boldsymbol{x}_0 \in \mathcal{B}$. For any $\alpha \in [0, 1]$, let

$$\boldsymbol{x}_\alpha \triangleq \alpha \boldsymbol{x}^* + (1 - \alpha)\boldsymbol{x}_0.$$

By convexity, we have $\boldsymbol{x}_\alpha \in \mathcal{B}$ for any $\alpha$. Note that $g(\widehat{\boldsymbol{x}}, \boldsymbol{x}_\alpha)$ is differentiable. By the definition of $x^*$, we have

$$(\boldsymbol{x}^* - \widehat{\boldsymbol{x}})^{\mathsf{T}} A(\boldsymbol{x}^* - \boldsymbol{x}_0) = \frac{1}{2}\frac{\partial}{\partial \alpha} g(\widehat{\boldsymbol{x}}, \boldsymbol{x}_\alpha)\Big|_{\alpha=1} \le 0.$$

Therefore,

$$g(\boldsymbol{x}^*, \boldsymbol{x}_0) = g(\widehat{\boldsymbol{x}}, \boldsymbol{x}_0) + 2(\boldsymbol{x}^* - \widehat{\boldsymbol{x}})^{\mathsf{T}} A(\boldsymbol{x}^* - \boldsymbol{x}_0) - g(\boldsymbol{x}^*, \widehat{\boldsymbol{x}}) \le g(\widehat{\boldsymbol{x}}, \boldsymbol{x}_0),$$

where the last inequality uses the PSD property of $A$.

Now we consider the more general case and let $\boldsymbol{x}$ be any vector in $\mathcal{B}$. Following the same steps in the earlier case, we have

$$(\boldsymbol{x}^* - \widehat{\boldsymbol{x}})^{\mathsf{T}} A(\boldsymbol{x}^* - \boldsymbol{x}) \le 0.$$

Hence,

$$\begin{aligned} g(\boldsymbol{x}^*, \boldsymbol{z}_0) - g(\widehat{\boldsymbol{x}}, \boldsymbol{z}_0) &= 2(\boldsymbol{x}^* - \widehat{\boldsymbol{x}})^{\mathsf{T}} A(\boldsymbol{x}^* - \boldsymbol{z}_0) - g(\boldsymbol{x}^*, \widehat{\boldsymbol{x}}) \\ &\le 2(\boldsymbol{x}^* - \widehat{\boldsymbol{x}})^{\mathsf{T}} A(\boldsymbol{x} - \boldsymbol{z}_0) - g(\boldsymbol{x}^*, \widehat{\boldsymbol{x}}) \\ &= g(\boldsymbol{z}_0, \boldsymbol{x}) - (\boldsymbol{x} - \boldsymbol{z}_0 - \boldsymbol{x}^* + \widehat{\boldsymbol{x}})^{\mathsf{T}} A(\boldsymbol{x} - \boldsymbol{z}_0 - \boldsymbol{x}^* + \widehat{\boldsymbol{x}}) \\ &\le g(\boldsymbol{z}_0, \boldsymbol{x}). \end{aligned}$$

Note that the above inequality holds for any $\boldsymbol{x} \in \mathcal{B}$. The proposition is proved by taking the minimum over $\boldsymbol{x}$.

$\square$

# B  Proof of Proposition 3.3

*Proof.* We first prove for the case where $\boldsymbol{Z}$ is deterministic. Let $\mu_{\boldsymbol{Z}}$ denote the conditional expectation of $\theta$. By Cauchy's inequality,

$$\mathbb{E}[(\theta - \mu_{\boldsymbol{Z}})^2 | \boldsymbol{Z}] \cdot \mathbb{E}\left[\left(\frac{\partial}{\partial \theta} \ln f_{\boldsymbol{Z}}(\theta)\right)^2 \Big| \boldsymbol{Z}\right] \ge \mathbb{E}\left[\left|(\theta - \mu_{\boldsymbol{Z}}) \cdot \frac{\partial}{\partial \theta} \ln f_{\boldsymbol{Z}}(\theta)\right| \Big| \boldsymbol{Z}\right]^2. \tag{22}$$

The quantity on the RHS above can be bounded as follows.

$$\mathbb{E}\left[\left|(\theta - \mu_{\mathbf{Z}}) \cdot \frac{\partial}{\partial \theta} \ln f_{\mathbf{Z}}(\theta)\right| \middle| \mathbf{Z}\right] = \int \left|(\theta - \mu_{\mathbf{Z}}) \cdot \frac{\partial}{\partial \theta} \ln f_{\mathbf{Z}}(\theta)\right| f_{\mathbf{Z}}(\theta) d\theta$$

$$= \int \left|(\theta - \mu_{\mathbf{Z}}) \cdot \frac{\partial}{\partial \theta} f_{\mathbf{Z}}(\theta)\right| d\theta$$

$$\geq \limsup_{T \to +\infty} \left|\int_{-T}^{T} (\theta - \mu_{\mathbf{Z}}) \cdot \frac{\partial}{\partial \theta} f_{\mathbf{Z}}(\theta) d\theta\right|$$

$$= \limsup_{T \to +\infty} \left|\left((\theta - \mu_{\mathbf{Z}}) f_{\mathbf{Z}}(\theta)\big|_{\theta=-T}^{\theta=T}\right) - \mathbb{P}[\theta \in [-T, T]|\mathbf{Z}]\right|$$

$$\geq 1,$$

where the last inequality uses the integrability of $f_{\mathbf{Z}}$, which implies

$$\liminf_{T \to +\infty} (\theta - \mu_{\mathbf{Z}}) f_{\mathbf{Z}}(\theta)\big|_{\theta=-T}^{\theta=T} \leq 0.$$

Then we evaluate the second factor on the LHS of inequality (22). Recall that $\frac{\partial^2}{\partial \theta^2} \ln f_{\mathbf{Z}}(\theta)$ is integrable, the following limit exists.

$$\mathbb{E}\left[\frac{\partial^2}{\partial \theta^2} \ln f_{\mathbf{Z}}(\theta)\middle| \mathbf{Z}\right] = \lim_{T \to +\infty} \int_{-T}^{T} f_{\mathbf{Z}}(\theta) \frac{\partial^2}{\partial \theta^2} \ln f_{\mathbf{Z}}(\theta) d\theta.$$

Then by positivity, we also have

$$\mathbb{E}\left[\left(\frac{\partial}{\partial \theta} \ln f_{\mathbf{Z}}(\theta)\right)^2 \middle| \mathbf{Z}\right] = \lim_{T \to +\infty} \int_{-T}^{T} f_{\mathbf{Z}}(\theta) \left(\frac{\partial}{\partial \theta} \ln f_{\mathbf{Z}}(\theta)\right)^2 d\theta.$$

If we focus the non-trivial case where the first limit is not $-\infty$, the above two equation implies the existence of the following limit.

$$\mathbb{E}\left[\frac{\partial^2}{\partial \theta^2} \ln f_{\mathbf{Z}}(\theta)\middle| \mathbf{Z}\right] + \mathbb{E}\left[\left(\frac{\partial}{\partial \theta} \ln f_{\mathbf{Z}}(\theta)\right)^2 \middle| \mathbf{Z}\right]$$

$$= \lim_{T \to +\infty} \int_{-T}^{T} f_{\mathbf{Z}}(\theta) \left(\frac{\partial^2}{\partial \theta^2} \ln f_{\mathbf{Z}}(\theta) + \left(\frac{\partial}{\partial \theta} \ln f_{\mathbf{Z}}(\theta)\right)^2\right) d\theta$$

$$= \lim_{T \to +\infty} f_{\mathbf{Z}}(\theta) \frac{\partial}{\partial \theta} \ln f_{\mathbf{Z}}(\theta) \bigg|_{\theta=-T}^{\theta=T}$$

$$= \lim_{T \to +\infty} \frac{\partial}{\partial \theta} f_{\mathbf{Z}}(\theta) \bigg|_{\theta=-T}^{\theta=T}.$$

The result of the above equation has to be zero, because the limit points of $\frac{\partial}{\partial \theta} f_{\mathbf{Z}}(\theta)$ must contain zero on both ends of the real line, which is implied by the integrability of $f_{\mathbf{Z}}$. Consequently, we have

$$\mathbb{E}\left[\left(\frac{\partial}{\partial \theta} \ln f_{\mathbf{Z}}(\theta)\right)^2 \middle| \mathbf{Z}\right] = \mathbb{E}\left[-\frac{\partial^2}{\partial \theta^2} \ln f_{\mathbf{Z}}(\theta)\middle| \mathbf{Z}\right]. \tag{23}$$

Then, the special case of Proposition 3.3 with fixed $\mathbf{Z}$ is implied by inequality (22).

When $\mathbf{Z}$ is variable, we simply have

$$\mathbb{E}\left[\mathrm{Var}[\theta|\mathbf{Z}]\right] \geq \mathbb{E}\left[1/\mathbb{E}\left[\left(\frac{\partial}{\partial \theta} \ln f_{\mathbf{Z}}(\theta)\right)^2 \middle| \mathbf{Z}\right]\right]$$

$$\geq \frac{1}{\mathbb{E}\left[\mathbb{E}\left[\left(\frac{\partial}{\partial \theta} \ln f_{\mathbf{Z}}(\theta)\right)^2 \middle| \mathbf{Z}\right]\right]}.$$

Then the proposition is implied by equation (23). $\qquad\square$

## C  Proof of Theorem 2.2

We first investigate the lower bounds. Observe that the proof provided in Section 3.1 only fails when the constructed hard instances have $||\boldsymbol{x}_0||_2 > 1$. Hence, we have already covered the $T \geq \left(\sum_{k=1}^{d} \lambda_k^{-\frac{1}{2}}\right)\left(\sum_{k=1}^{d} \lambda_k^{-\frac{3}{2}}\right)$ case, i.e., when $k^* = \dim A = d$. It remains to consider the other scenarios, where $k^* < d$ is satisfied.

By the assumption that $T \geq \left(\sum_{k=1}^{k^*} \lambda_k^{-\frac{1}{2}}\right)\left(\sum_{k=1}^{k^*} \lambda_k^{-\frac{3}{2}}\right)$, one can instead set the entries of $\boldsymbol{x}_0$ in the earlier proof with indices greater than $k^*$ to be zero, so that $||\boldsymbol{x}_0||_2 \leq 1$ is satisfied. Formally, let the hard-instance functions be constructed by the following set.

$$\boldsymbol{x}_0 \in \mathcal{X}_{\mathrm{H}} \triangleq \left\{ (x_1, x_2, ..., x_{k^*}, 0, ..., 0) \;\middle|\; x_k = \pm\sqrt{\frac{\lambda_k^{-\frac{3}{2}}\left(\sum_j \lambda_j^{-\frac{1}{2}}\right)}{2T}}, \forall k \in [k^*] \right\}.$$

Then by the identical proof steps, we have $\mathfrak{R}(T; A) = \Omega\left(\left(\sum_{k=1}^{k^*} \lambda_k^{-\frac{1}{2}}\right)^2 \Big/ T\right)$.

Next, we show that $\mathfrak{R}(T; A) = \Omega\left(\lambda_{k^*+1}\right)$. We assume the non-trivial case where $\lambda_{k^*+1} \neq 0$. Note that $\mathfrak{R}(T; A)$ is non-increasing w.r.t. $T$. We can lower bound $\mathfrak{R}(T; A)$ through the above steps but by replacing $T$ with any larger quantity. Specifically, recall that $k^*$ is largest integer satisfying $T \geq \left(\sum_{k=1}^{k^*} \lambda_k^{-\frac{1}{2}}\right)\left(\sum_{k=1}^{k^*} \lambda_k^{-\frac{3}{2}}\right)$, which implies $T \leq \left(\sum_{k=1}^{k^*+1} \lambda_k^{-\frac{1}{2}}\right)\left(\sum_{k=1}^{k^*+1} \lambda_k^{-\frac{3}{2}}\right)$. We have,

$$\mathfrak{R}(T; A) \geq \mathfrak{R}\left(\left(\sum_{k=1}^{k^*+1} \lambda_k^{-\frac{1}{2}}\right)\left(\sum_{k=1}^{k^*+1} \lambda_k^{-\frac{3}{2}}\right); A\right).$$

Notice that this change of sampling time allows us to apply the earlier lower bound with $k^*$ incremented by 1.

$$\mathfrak{R}(T; A) \geq \Omega\left(\frac{\left(\sum_{k=1}^{k^*+1} \lambda_k^{-\frac{1}{2}}\right)^2}{\left(\sum_{k=1}^{k^*+1} \lambda_k^{-\frac{1}{2}}\right)\left(\sum_{k=1}^{k^*+1} \lambda_k^{-\frac{3}{2}}\right)}\right)$$

$$= \Omega\left(\frac{\sum_{k=1}^{k^*+1} \lambda_k^{-\frac{1}{2}}}{\sum_{k=1}^{k^*+1} \lambda_k^{-\frac{3}{2}}}\right) = \Omega(\lambda_{k^*+1}).$$

To conclude,

$$\mathfrak{R}(T; A) = \Omega\left(\max\left\{\frac{\left(\sum_{k=1}^{k^*} \lambda_k^{-\frac{1}{2}}\right)^2}{T}, \lambda_{k^*+1}\right\}\right) = \Omega\left(\frac{\left(\sum_{k=1}^{k^*} \lambda^{-\frac{1}{2}}\right)^2}{T} + \lambda_{k^*+1}\right),$$

which completes the proof of the lower bounds.

The needed upper bounds can be obtained by only estimating the first $k^*$ entries of $\boldsymbol{x}_0$.

**Remark C.1.** *The requirement of $T > 3\dim A$ in the Theorem statement is simply due to the integer constraints for the achievability bounds. Indeed, when $\lambda_{\dim A}$ is large, it requires at least $\Omega(\dim A)$ samples to achieve $O(1)$ expected simple regret.*

## D  Proof Details for Theorem 2.4

### D.1  Truncation Method and Its Applications

The truncation method is based on the following facts.

**Proposition D.1.** *For any sequence of independent random variables $X_1, X_2, ..., X_n$ and any fixed parameter $m$ satisfying $m > \max_k |\mathbb{E}[X_k]|$. Let $Z_k = \max\{\min\{X_k, m\}, -m\}$ for any $k \in [n]$, we have*

$$|\mathbb{E}[Z_k] - \mathbb{E}[X_k]| \leq \frac{1}{4} \cdot \frac{\text{Var}[X_k]}{m - |\mathbb{E}[X_k]|}, \tag{24}$$

$$\text{Var}[Z_k] \leq \mathbb{E}\left[(Z_k - \mathbb{E}[X_k])^2\right] \leq \text{Var}[X_k]. \tag{25}$$

*Moreover, for any $z > 0$, we have*

$$\mathbb{P}\left[\left|\sum_k Z_k - \sum_k \mathbb{E}[X_k]\right| \geq z\right] \leq 2\exp\left(\sum_k \frac{\text{Var}[X_k]}{m(m - |\mathbb{E}[X_k]|)} - \frac{z}{m}\right). \tag{26}$$

*Proof.* The first inequality is proved by expressing the LHS with piecewise linear functions. Note that by the definition of $Z_k$, we have

$$\begin{aligned}|\mathbb{E}[Z_k] - \mathbb{E}[X_k]| &= |\mathbb{E}[\max\{-m - X_k, 0\}] - \mathbb{E}[\max\{X_k - m, 0\}]| \\ &\leq |\mathbb{E}[\max\{-m - X_k, 0\}]| + |\mathbb{E}[\max\{X_k - m, 0\}]| \\ &= \mathbb{E}[\max\{|X_k| - m, 0\}].\end{aligned}$$

We apply the following inequalities, which holds for any $m \geq |\mathbb{E}[X_k]|$.

$$|X_k| - m \leq |X_k - \mathbb{E}[X_k]| - m + \mathbb{E}[X_k] \leq \frac{1}{4} \cdot \frac{|X_k - \mathbb{E}[X_k]|^2}{m - \mathbb{E}[X_k]}.$$

Therefore,

$$\begin{aligned}|\mathbb{E}[Z_k] - \mathbb{E}[X_k]| &\leq \mathbb{E}\left[\frac{1}{4} \cdot \frac{|X_k - \mathbb{E}[X_k]|^2}{m - \mathbb{E}[X_k]}\right] \\ &= \frac{1}{4} \cdot \frac{\text{Var}[X_k]}{m - |\mathbb{E}[X_k]|}.\end{aligned}$$

The second inequality is due to the following elementary facts,

$$\mathbb{E}[(Z_k - \mathbb{E}[X_k])^2] \leq \mathbb{E}[(X_k - \mathbb{E}[X_k])^2] = \text{Var}[X_k],$$

where the inequality step is implied by the definition of $Z_k$ and the condition $m > \max_k |\mathbb{E}[X_k]|$.

To prove the third inequality, we first investigate the following upper bound, which is due to Markov's inequality.

$$\begin{aligned}\mathbb{P}\left[\sum_k Z_k - \sum_k \mathbb{E}[X_k] \geq z\right] &\leq \frac{\mathbb{E}[e^{\frac{1}{m}(\sum_k Z_k - \sum_k \mathbb{E}[X_k])}]}{e^{\frac{z}{m}}} \\ &= \frac{\prod_k \mathbb{E}[e^{\frac{1}{m}(Z_k - \mathbb{E}[X_k])}]}{e^{\frac{z}{m}}}\end{aligned} \tag{27}$$

The equality step above is by the fact that $Z_k$'s are jointly independent. For each $k$, using the fact that $Z_k$ is bounded, particularly, $Z_k - \mathbb{E}[X_k] \leq m + |\mathbb{E}[X_k]|$, we have the following inequality

$$e^{\frac{1}{m}(Z_k - \mathbb{E}[X_k])} - 1 - \frac{1}{m}(Z_k - \mathbb{E}[X_k]) \leq (Z_k - \mathbb{E}[X_k])^2 \cdot \frac{e^{\frac{1}{m}(m + |\mathbb{E}[X_k]|)} - 1 - \frac{1}{m}(m + |\mathbb{E}[X_k]|)}{(m + |\mathbb{E}[X_k]|)^2}.$$

For brevity, let $\theta \triangleq \frac{|\mathbb{E}[X_k]|}{m}$. We combine the above bound with inequality (24) and (25) to obtain that

$$\begin{aligned}\mathbb{E}[e^{\frac{1}{m}(Z_k - \mathbb{E}[X_k])}] &= 1 + \mathbb{E}\left[\frac{1}{m}(Z_k - \mathbb{E}[X_k])\right] + \mathbb{E}\left[e^{\frac{1}{m}(Z_k - \mathbb{E}[X_k])} - 1 - \frac{1}{m}(Z_k - \mathbb{E}[X_k])\right] \\ &\leq 1 + \frac{\text{Var}[X_k]}{m(m - |\mathbb{E}[X_k]|)} \cdot \left(\frac{1}{4} + (1 - \theta) \cdot \frac{e^{1+\theta} - 2 - \theta}{(1 + \theta)^2}\right).\end{aligned} \tag{28}$$

Recall that $\theta < 1$ as assumed in the proposion. From elementary calculus, we have

$$\mathbb{E}[e^{\frac{1}{m}(Z_k - \mathbb{E}[X_k])}] \leq 1 + \frac{\mathrm{Var}[X_k]}{m(m - |\mathbb{E}[X_k]|)}$$

$$\leq \exp\left(\frac{\mathrm{Var}[X_k]}{m(m - |\mathbb{E}[X_k]|)}\right).$$

Therefore, recall inequality (27), we have

$$\mathbb{P}\left[\sum_k Z_k - \sum_k \mathbb{E}[X_k] \geq z\right] \leq \exp\left(\sum_k \frac{\mathrm{Var}[X_k]}{m(m - |\mathbb{E}[X_k]|)} - \frac{z}{m}\right).$$

By symmetry, one can also prove the following bound through the same steps.

$$\mathbb{P}\left[\sum_k Z_k - \sum_k \mathbb{E}[X_k] \leq -z\right] \leq \exp\left(\sum_k \frac{\mathrm{Var}[X_k]}{m(m - |\mathbb{E}[X_k]|)} - \frac{z}{m}\right).$$

Hence, the needed inequality is obtained by adding the two inequalities above. $\qquad\square$

Now equation (14) for the 1D case is immediately implied by Proposition D.1. Recall the construction of $\widehat{A}$ in the proof, for any sufficiently large $T_0$, we have

$$\mathbb{P}\left[\left|\widehat{A} - A\right| \geq T_0^{-\alpha}\right] \leq 2\exp\left(3 - \frac{T_0^{0.5 - \alpha}}{3}\right) = o\left(\frac{1}{T_0^{\beta}}\right).$$

**Remark D.2.** *Instead of projecting to a bounded interval, the same achievability result can be obtained if the we average over any functions that map the samples to $[-T_0^{0.5}, T_0^{0.5}]$ while imposing an additional error of $o(T^{-\alpha})$ everywhere. This includes $\Theta(\ln T)$-bit uniform quantizers, which naturally appear in digital systems, over which exact computation can be performed to eliminate numerical errors. We present this simple generalization in the following corollary.*

**Corollary D.3.** *Consider the setting in Proposition D.1. Let $Y_1, ..., Y_n$ be variables that satisfy $|Y_k - Z_k| \leq b$ for all $k$ with probability 1. We have*

$$\mathbb{P}\left[\left|\sum_k Y_k - \sum_k \mathbb{E}[X_k]\right| \geq z\right] \leq 2\exp\left(\sum_k \frac{\mathrm{Var}[X_k]}{m(m - |\mathbb{E}[X_k]|)} - \frac{z - bn}{m}\right).$$

## D.2 Proof of Proposition 4.2

*Proof.*

$$\boldsymbol{y}^\mathsf{T} Z \boldsymbol{y} - \boldsymbol{y}^\mathsf{T} \widehat{A}_0 \boldsymbol{y} = \boldsymbol{y}^\mathsf{T}(Z - \widehat{A}_0)\boldsymbol{y} \leq ||Z - \widehat{A}_0||_\mathrm{F} ||\boldsymbol{y}||_2^2 \leq ||Z - \widehat{A}_0||_\mathrm{F} ||\widehat{A}_0^{-1}||_\mathrm{F} (\boldsymbol{y}^\mathsf{T} \widehat{A}_0 \boldsymbol{y}).$$

$\qquad\square$

## D.3 Proof of inequality (19)

We apply Proposition 4.2 to inequality (17) and let $Z = A\widehat{A}_0^{-1}A$. Note that

$$||Z - \widehat{A}_0||_\mathrm{F} \leq 2||A - \widehat{A}_0||_\mathrm{F} + ||(A - \widehat{A}_0)\widehat{A}_0^{-1}(A - \widehat{A}_0)||_\mathrm{F} = o(1),$$

which satisfies the condition of Proposition 4.2. Using the fact that $\widehat{A}_0^{-1} \widehat{A}_0 \widehat{A}_0^{-1} = \widehat{A}_0^{-1}$, we have

$$\mathbb{E}\left[\left(\widehat{A}_0^{-1}A\left(\widehat{\boldsymbol{x}} - \widehat{A}_0^{-1}A\boldsymbol{x}_0\right)\right)^\mathsf{T} \widehat{A}_0 \left(\widehat{A}_0^{-1}A\left(\widehat{\boldsymbol{x}} - \widehat{A}_0^{-1}A\boldsymbol{x}_0\right)\right)\right]$$

$$= \mathbb{E}\left[\left(\widehat{\boldsymbol{x}} - \widehat{A}_0^{-1}A\boldsymbol{x}_0\right)^\mathsf{T} Z \left(\widehat{\boldsymbol{x}} - \widehat{A}_0^{-1}A\boldsymbol{x}_0\right)\right] = O\left((\mathrm{Tr}(A^{-\frac{1}{2}}))^2\right)/T. \qquad (29)$$

Then by the triangle inequality for the PSD matrix $\widehat{A}_0$, the combination of the above inequality and inequality (18) gives

$$\mathbb{E}\left[(\widetilde{\boldsymbol{x}}_T - \boldsymbol{z})^\mathsf{T} \widehat{A}_0 (\widetilde{\boldsymbol{x}}_T - \boldsymbol{z})\right] = O\left((\mathrm{Tr}(A^{-\frac{1}{2}}))^2\right)/T.$$

## D.4 Proof of Proposition 4.3

*Proof.* When $A_1$ and $A$ has the same rank, the map $P_1$ is invertible over the column space of $A$. Under such condition, there exists a matrix $X$ such that $A = X P_1 A$. Note that $A_1 A_1^{-1} = P_1$. We have $X P_1 = X P_1 A_1 A_1^{-1} = A A_1^{-1}$. Therefore, the needed $A = A A_1^{-1} A$ is obtained by multiplying $A$ on the right-hand sides in the above identity. $\square$

