# OpenReview forum: "Sample Complexity for Quadratic Bandits: Hessian Dependent Bounds and Optimal Algorithms"
_NeurIPS.cc/2023/Conference — NeurIPS 2023 poster_

### Official Review · Reviewer_gm1W · 2023-06-29

**Soundness:** 3 good
**Presentation:** 1 poor
**Contribution:** 3 good
**Rating:** 6
**Confidence:** 3

**Summary:**

The paper considers the problem of Quadratic Hessian bandit and establishes both lower and upper bounds on algorithm performance to provide a characterization of simple regret in this setting. The authors also propose an algorithm that is agnostic to the knowledge of the Hessian.

**Strengths:**

The paper considers a new problem on quadratic bandits and provides a complete characterization of simple regret in the setting. The completeness and tightness of the results paints a clear picture of the results about this new problem.

**Weaknesses:**

One thing that the paper needs to improve upon is the writing. The exposition lacks a flow and proofs of the theorems seem to be all over the place.  Some points that I realized can be improved:
1. The proof of Theorem 2.1 can be improved. It took me quite a bit of effort to understand the steps mainly because some intermediate steps were missing. The missing steps were non-trivial in the sense if they are missing, it is not easy for the reader to make the connection unless they have several results handy. I would suggest to please ensure that the proof is largely self-contained and it can be understood by the reader without actually having them to do intermediate steps. If it cannot be accommodated in the main text, you can move some intermediate result to Appendix or atleast provide a citation of some the results being used in the intermediate steps e.g. MMSE estimator. Also I think you are missing an "$L_k$" in equation 7.
2. The statement of Theorem 2.1 is not clear. From the text leading upto it and the proof, it seems that it is a statement about the lower bound. But there is no lower bound anywhere in the statement of the theorem (apart from the one that is implicit in the limit for Gaussian noise). I would suggest to please make the statement of theorem clear and precise.
3. Similarly, the way Theorem 2.4 is phrased, it is (almost) immediate from the current statement of Theorem 2.1. I think in addition to universal optimality you also need to mention that this algorithm does not need the prior information about the Hessian. This not only makes the theorem clearer but also stronger.

In addition to the above comments, I encourage the authors to please check the exposition for consistency and flow.

**Questions:**

The choice of $x_k$ and the lower bound on $T$ in Theorem 2.2. (line 67) are rather interesting. Do you have an intuitive/physical explanation of why that particular expression comes up or what that represents? I haven't come across that combination of $\lambda_k^{-3/2}$ and $\lambda^{-1/2}$ anywhere else and it might have something fundamentally insightful.

**Limitations:**

Yes.

---

> ### Author Rebuttal · Authors · 2023-08-09
>
> Thank you for the suggestions and below are the detailed responses.
>
> Q1: Details for the Proof of Theorem 2.1.
>
> A1: We will provide additional details for the intermediate steps, including additing a reference for the optimality of MMSE estimators. We will also correct the typo in equation (7).
>
> Q2: Clarity on the statement of Theorem 2.1.
>
> A2: We would first like to clarify that Theorem 2.1 contains both upper bounds and lower bounds, only that the proof of the upper bound is much simpler. We decided that the upper bound is to be included as it can be compared to Theorem 2.4. However, we can better emphasis the lower bound by revising the last sentence to be “..., i.e., a matching lower bound implies the following equality”.
>
> Q3: Clarity on the statement of Theorem 2.4.
>
> A3: Indeed, in Theorem 2.4 we are exactly providing an algorithm that is independent of the Hessian (which turns out to be non-trivial compared to its counterpart in Theorem 2.1). We will revise the statement to “There exists an algorithm $\mathcal{A}$, which does not depend on the Hessian parameter $A$, such that for $A$ being any PSD matrix, the achieved minimax…” to better clarify this fact.
>
> Q4: Choice of $x_k$ and the transition threshold of $T$.
>
> A4: Intuitively $x_k$ is selected such that the sample complexity for learning the sign of $x_k$ is proportional to the penalty of an incorrectly estimation, so that the performance of an algorithm on these special cases is roughly independent of how many samples are used in each direction. The explicit exponents ($1/2$ and $3/2$) is a result of the quadratic growth of the objective fuction. We believe that if the function class is instead in the form of $f=\sum \lambda_k (x_k-x_{0,k})^{\alpha}$ for some general $\alpha$, then the correct $|x_k|$ for the hard instances will be proportional to $\lambda_{k}^{-\frac{3}{\alpha+2}}$.

---

> > ### Comment · Reviewer_gm1W · 2023-08-14
> > **Response to Rebuttal**
> >
> > Thank you for your response. I will maintain my score and suggest the authors to edit the manuscript based on my earlier comments should it go through. I have no further questions.

---

### Official Review · Reviewer_Cwak · 2023-07-04

**Soundness:** 3 good
**Presentation:** 3 good
**Contribution:** 3 good
**Rating:** 7
**Confidence:** 3

**Summary:**

This papers studies quadratic stochastic zeroth-order optimization and provide lower and upper bounds on the minimax simple regret. First, the authors provide an asymptotically tight upper bound of $\frac{1}{2}({\rm Tr}(A^{-1/2}))^2$ where $A$ is the Hessian of the quadratic. Then, the authors provide asymptotic lower bounds which take two distinct expressions depending on properties of $A$. Next, the authors show the existence of a universally optimal algorithm, meaning that it asymptotically achieves the Hessian-dependent bounds without accessing the Hessian information.

**Strengths:**

Clear problem statement and result presentation.

**Weaknesses:**

See **Questions**.

**Questions:**

- "Energy allocation" seems to be the same as "energy spectrum" defined in Section 3.1. Please formally define energy allocation or unify the terminology.
- Line 207: serve --> served.
- Line 195: remove "an".
- Please formally define Truncated Diff (projecting the diff onto [-t, t]) and other simple operations in the algorithms (Simple Project, etc.).
- Regarding Section 4.1, are there potential ways to do better than applying the 1D estimator poly(d) times to obtain an estimate for $A$? Are there more efficient join estimators that make use of $A$ being p.s.d.? If so, it would be helpful to point out (without incorporating them in this paper's analysis as it does not seem necessary). It might still be useful in future finite-sample analysis.

**Limitations:**

The authors defined the scope of this paper to be within specific problem setups. This work is largely theoretical and does not have potential negative societal impact.

---

> ### Author Rebuttal · Authors · 2023-08-09
>
> We would like to thank the reviewer for their comments, and will revise the identified typos. Below are the point-to-point responses to the comments.
>
> Q1: Energy allocation vs energy spectrum
>
> A1: Energy spectrum refers to the particular function defined above line 86, and we used energy allocation to refer to the act of designing the algorithm to match the energy spectrum to any particular vector (e.g., assign $2t_k$ samples to the $k$th dimension in Algorithm 1). This will be better clarified in the revised version.
>
> Q2: Formal definition of subroutines
>
> A2: Thanks for the suggestion, we will add the formal definitions in the corresponding algorithm enviromennts. For example, we will revise the pseudo code of the truncated diff as follows:
>
>   - For $k\leftarrow  1 $ to $t$
>
>     + Let $y_+$, $y_-$ each be a sample of $f$ at $x_0$, $x_1$, respectively
>     + Compute the projection of the difference $y_+-y_-$ to the interval $[-t^{0.5}, t^{0.5}]$, i.e., let $z_k=...$
>   - End for
>
>   - Return the average $\frac{1}{t} \sum_{k=1}^t z_k $
>
> Q3: Better ways to estimate the Hessian?
>
> A3: Instead of estimating the Hessian entrywise, there are several possible alternatives, e.g., see  the following paper [A]. However, for quadratic f, Hessian estimation entrywise is order-optimal (in terms of squared Frobenius norm of the error): As a lower bound, learning the hessian is harder than learning a vector length $d(d+1)/2$ by observing its unit projections under Gaussian noise (because sampling any $f=x^{\intercal}Ax$ essentially returns a noisy version of a projection of its $n(n+1)/2$ independent entries), hence we need at least $\Omega(d^4/\epsilon^2)$ samples to achieve a squared estimation error of $\epsilon^2$. This exactly matches the error achieve by entry-wise estimation. For the same reason, the PSD property will not reduce the optimal sample complexity beyond a constant factor.
>
> [A] Balasubramanian and Ghadimi, “Zeroth-order Nonconvex Stochastic Optimization: Handling Constraints, High-Dimensionality and Saddle-Points”

---

> > ### Comment · Reviewer_Cwak · 2023-08-18
> >
> > Thanks for the explanation on the complexity of Hessian estimation, truncated diff steps and energy terminologies. They look good to me.

---

### Official Review · Reviewer_q1aQ · 2023-07-05

**Soundness:** 3 good
**Presentation:** 3 good
**Contribution:** 2 fair
**Rating:** 4
**Confidence:** 3

**Summary:**

The paper considers the black-box optimization problem of quadratic functions specified with PSD matrices. The paper shows matching upper and lower sample complexity bounds depending on the matrices. For the upper bound, the paper proposes algorithms.

**Strengths:**

The paper gives tight upper and lower sample complexity bounds for the black-box optimization of quadratic functions. The strength of the result is that the bounds depend on the PSD matrix, which means the bounds are instance-dependent. These results recovers previous results as well.

**Weaknesses:**

I feel that the focus of the paper is a bit narrow. So far, I do not know any motivating situation where we need to solve some black-box optimization of quadratic functions. Although the results are technically solid, their motivation and practical usefulness are not strong enough.
The paper would be better evaluated for more theory/optimization-oriented conferences/journals.

Or, if the proposed algorithms were more practical, say, it works for general convex functions but works better when the function is quadratic (best-of-both-world type algorithms), I would evaluate the paper higher.

**Questions:**

Do the proposed algorithm work for general convex functions?


I read other reviews and rebuttal comments. The rebuttal answers my question sufficiently. But I would keep my score the same since my concern about the motivation is not resolved.

**Limitations:**

As raised above, the technical results are solid, but practical usefulness is limited.

---

> ### Author Rebuttal · Authors · 2023-08-09
>
> Thank you for the feedback. Convex quadratic functions depict the local geometry of strongly convex functions near the critical point, and their sample complexity remains an open question in the online learning community. Although our result does not directly apply to general convex functions. Fully characterizing the sample complexity in this important (quadratic) case can serve as a stepping stone that provides essential techniques and a better understanding of how to achieve optimal sample complexities in general cases.

---

### Official Review · Reviewer_tYpf · 2023-07-09

**Soundness:** 2 fair
**Presentation:** 3 good
**Contribution:** 3 good
**Rating:** 5
**Confidence:** 4

**Summary:**

This paper studies zero-th order quadratic optimization with bandit feedback. It provides a comprehensive analysis of the optimal sample complexity, which depends on the Hessian of the objective function. There are two contributions. Firstly, it establishes lower bounds on Hessian-dependent complexities using the concept of energy allocation, capturing the interaction between the search algorithm and the geometry of the problem. The matching upper bound is achieved through optimizing the energy spectrum. Secondly, an algorithm is presented that achieves asymptotically optimal sample complexities for all Hessian instances, independent of the Hessian. These optimal sample complexities remain valid even for heavy-tailed additive noise distributions, enabled by a truncation method.

**Strengths:**

1. If the result is correct, it provides matching upper and lower bounds for quadratic bandits, which is pretty nice

2. The results hold even for heavy-tailed noise with the help of truncation.

**Weaknesses:**

1. The definition of the effective dimension $k^*$ does not look correct

2. The proposed algorithm needs to know a lot of problem-related quantities, which is not practical

3. There is no experimental evaluation

**Questions:**

What is $dimA$? It is not defined. Is it just $d$?

In Line 67, in the definition of the effective dimension $k^*$, do you miss a fractional number notion? i.e., $/$ Otherwise, it would be an ill-defined dimension.

The construction of the lower bound is a bit strange. When you construct x_0 in this way, it will naturally lead to the definition of the effective dim $k^*$. I’m wondering if the current lower bound is truly essential, or just an artifact?

In my opinion, the non-asymptotic lower bound (Theorem 2.2) is more interesting, why don’t you present it instead of proving a weaker (and asymptotic) lower bound in the main paper?

**Limitations:**

The zero-th order quadratic optimization problem is a bit simple. But it is a good starting point

The lower bound proof technique lacks generality. It might be simplified by the generalized le cam’s method. See the textbook “Bandit algorithms”

---

> ### Author Rebuttal · Authors · 2023-08-09
>
> Thank you for the comments. We have now provided the following point-to-point response.
>
> Q1: The definition of dim$A$?
>
> A1: In the submitted version, our result is stated for dim$A=d$ where dim refers to the dimension of the Euclidian space over which A is defined. While our result holds true under this definition, it later came to our awareness that Theorem 2.2 still holds if dim$A$ instead denotes the rank of $A$, and the proof follows almost the same arguments. We plan to let dimA denote the rank of A in the revised version and update the proof steps accordingly.
>
> Q2: The definition of $k^*$ in line 67?
>
> A2:  We would like to verify that the condition of T in line 67 is correct under the convention of $0^{-1}=+\infty$. So, $k^*$ is always no greater than the rank of $A$. As we plan to update the definition of dim$A$ to be the rank of $A$, this clarity issue is automatically resolved as $\lambda_k^{-\frac{1}{2}}$ for $k\in\{1,2,...,\text{dim} A\}$ is always finite.
>
> Q3: Is the current lower bound truly essential or just an artifact?
>
> A3: First, we would like to clarify that the provided bounds themselves are fundamental as they are matched by upper bounds, and the choice of $k^*$ only affects how the bounds are presented. The transition threshold of T in Theorem 2.2 (or the definition of $k^*$) is fundamental to some degree, as in certain regimes, this specific choice of $k^*$ is required for the bound to be order-wise tight. There could be potential flexibility in how $k^*$ is selected for certain cases, but we choose to present one concrete example in our theorem for brevity.
>
> Q4: Why prove a weaker (and asymptotic) result in the main paper instead of the non-asymptotic lower bound (Theorem 2.2).
>
> A4: First, we would like to clarify that Theorem 2.1 is not merely a strictly weaker version of Theorem 2.2 as it characterizes the exact constant factor for the minimax asymptotic regret. Besides, the main paper focus on the proof for the asymptotic case for the following reasons: 1. The proof of Theorem 2.1 contains all essential ingredients (Theorem 2.2 is proved following exactly the same construction ideas, but with additional technical details in the analysis); 2. the statement of the theorem 2.1 is simpler compared to the non-asymptotic case, so the readers are not lost in technical details; 3. Theorem 2.1 is essential for understanding the significance of Theorem 2.4.
>
> Q5: Comparison to le cam’s method.
>
> A5: We would like to thank the reviewer for pointing out the reference. Compared to classical methods, such as le cam’s method or general lower-bounding approaches for bandit algorithms, a key distinction in our proof is to construct hard instances and provide analysis tailored for each Hessian instance. To the best of our knowledge, our proof (especially the analysis in Section 3.3) may not be directly obtained from earlier approaches, and we believe our proposed techniques can be a stepping stone to be applied for more general settings.

---

> > ### Comment · Reviewer_tYpf · 2023-08-18
> >
> > Thank you for your response. The authors have partly addressed my concern, regarding dimA and $k^*$. I am raising my score by +1 accordingly.

---

### Decision · Program_Chairs · 2023-09-21

**Decision:**

Accept (poster)

**Comment:**

This paper considers the problem of instance-dependent zeroth-order (bandit) convex optimization. Focusing on quadratic objectives, the main result of the paper is to provide algorithms that optimally adapt to the geometry of the problem instance under consideration via the Hessian, which they characterize via matching upper and lower bounds.

Reviewers agreed on the importance of developing an understanding of instance-dependent guarantees for zeroth-order optimization. They found the paper's algorithms and proof techniques to be novel, and agreed on the significance of characterizing the optimal hessian-dependent problem complexity. While some reviewers raised concerns around practicality and the restriction to quadratic objectives, they did not defend this position in discussion, and I believe it is appropriate to leave these issues for future research given the challenging nature of the problem.

For the final version of the paper, the authors are encouraged to incorporate the clarifications suggested by the reviewers.